# Quantitative MRI to Characterize Hypoxic Tumors in Comparison to FMISO PET/CT for Radiotherapy in Oropharynx Cancers

**DOI:** 10.3390/cancers15061918

**Published:** 2023-03-22

**Authors:** Pierrick Gouel, Françoise Callonnec, Franchel-Raïs Obongo-Anga, Pierre Bohn, Emilie Lévêque, David Gensanne, Sébastien Hapdey, Romain Modzelewski, Pierre Vera, Sébastien Thureau

**Affiliations:** 1Department of Radiology and Nuclear Medicine, Henri Becquerel Cancer Center and Rouen University Hospital, & QuantIF—LITIS [EA (Equipe d’Accueil) 4108–FR CNRS 3638], Faculty of Medicine, University of Rouen, 76000 Rouen, France; 2Department of Surgery, Henri Becquerel Cancer Center and Rouen University Hospital, 76000 Rouen, France; 3Unit of Clinical Reasearch, Henri Becquerel Cancer Center and Rouen University Hospital, 76000 Rouen, France; 4Department of Radiation Oncology, Henri Becquerel Cancer Center and Rouen University Hospital, & QuantIF—LITIS [EA (Equipe d’Accueil) 4108], 76000 Rouen, France

**Keywords:** hypoxia, head and neck cancer, FMISO PET, quantitative MRI, DCE-MRI, diffusion MRI, T1 mapping, T2 mapping, FDG PET, dose painting

## Abstract

**Simple Summary:**

The definition of tumor hypoxia is important in oncology because this characteristic is linked to a poor prognosis but remains debated because there are no reference modalities. In this context, we compared PET hypoxia (FMISO) and MRI data before surgery to determine the hypoxic volume at which to increase the radiotherapy dose in head and neck cancers. To our knowledge, this is the first study showing the value of combining tumor volumes obtained via PET and MRI to define the hypoxic lesion subvolume. The quantitative MRI parameters ADC, T1 mapping, and T2 mapping showed differences between hypoxic and normoxic volumes.

**Abstract:**

Intratumoral hypoxia is associated with a poor prognosis and poor response to treatment in head and neck cancers. Its identification would allow for increasing the radiation dose to hypoxic tumor subvolumes. 18F-FMISO PET imaging is the gold standard; however, quantitative multiparametric MRI could show the presence of intratumoral hypoxia. Thus, 16 patients were prospectively included and underwent 18F-FDG PET/CT, 18F-FMISO PET/CT, and multiparametric quantitative MRI (DCE, diffusion and relaxometry T1 and T2 techniques) in the same position before treatment. PET and MRI sub-volumes were segmented and classified as hypoxic or non-hypoxic volumes to compare quantitative MRI parameters between normoxic and hypoxic volumes. In total, 13 patients had hypoxic lesions. The Dice, Jaccard, and overlap fraction similarity indices were 0.43, 0.28, and 0.71, respectively, between the FDG PET and MRI-measured lesion volumes, showing that the FDG PET tumor volume is partially contained within the MRI tumor volume. The results showed significant differences in the parameters of SUV in FDG and FMISO PET between patients with and without measurable hypoxic lesions. The quantitative MRI parameters of ADC, T1 max mapping and T2 max mapping were different between hypoxic and normoxic subvolumes. Quantitative MRI, based on free water diffusion and T1 and T2 mapping, seems to be able to identify intra-tumoral hypoxic sub-volumes for additional radiotherapy doses.

## 1. Introduction

Definitive radiotherapy combined with concurrent chemotherapy is the contemporary standard of care in the nonsurgical management or adjuvant treatment of advanced stage head and neck squamous cell cancer (HNSCC) and is associated with variable 5-year disease-related outcomes ranging from 40 to 65%. Hypoxia is a major risk factor for local relapse or distant metastasis, especially in head and neck cancer treated with radiotherapy [1]. Recent data confirm the interest in personalized treatment according to tumor hypoxia data [2]. Indeed, in the case of the absence of hypoxic lesions, a therapeutic de-escalation could be considered. On the other hand, in the case of hypoxia, the risk of recurrence is important, particularly in regard to local recurrence, and therapeutic optimization is necessary.

Today, PET scans remain the reference examination to define hypoxic patients and areas that can benefit from therapeutic escalation [3]. Several tracers have been tested, but fluoromisodinazole (FMISO) remains the reference tracer despite having a low tumor-to-background signal [4,5]. A better definition of the hypoxic volume should allow considering an adapted treatment. An escalation in tumor subvolumes is technically feasible as intensity-modulated radiotherapy (IMRT) and adaptive radiotherapy make it possible to envisage personalized treatments based on treatment responses [6,7,8].

However, the definition of hypoxic volume remains debated, and there are no reference modalities [4,9,10]. Indeed, its correlation with the analysis of tumor volume in anatomopathology is complex, and many tracers have been tested according to variable modalities. As a complement to PET data, MRI may be an interesting examination method to define the hypoxic volume and could provide additional data. In addition to diagnostic MRI sequences, several other techniques have been proposed in the literature whose voxel intensity change would reflect tumor hypoxia, such as free water diffusion [11] or the use of a paramagnetic contrast medium [12], as well as measurements of T1 and T2 mappings to characterize the tissues [13,14]. However, the concomitant analysis of hypoxia PET and MRI data is very unusual for head and neck cancer [11,15].

The objective of this work is to define, from both PET hypoxia (FMISO) and MRI data, the hypoxic volume at which it would be of interest to increase the radiotherapy dose in head and neck cancers.

## 2. Materials and Method

### 2.1. Study

A total of 16 patients histologically diagnosed with squamous cell carcinoma of the oropharynx (operable primary tumor and measurable according to RECIST 1.1 evaluation criteria) were prospectively included (RTEP8-HYPONECK; NCT04031534). Each patient underwent Fluoro-2-Deoxy-D-Glucose (FDG) PET and FMISO PET-MRI before surgery; see Table 1.

### 2.2. FDG PET Imaging

FDG PET-CT whole-body images were acquired on a GE710 PET-CT device (General Electric, Milwaukee, WI, USA) 60 ± 5 min after the injection of approximately 3.5 MBq/kg of FDG. The PET acquisition time was 2 min per bed position. Between the intravenous injection of FDG and TEP acquisition, patients rested in quiet waiting rooms. PET and CT acquisition parameters were adapted to each patient. For patients with a body mass index (BMI) < 30 kg/m^2^, the corresponding X-ray computed tomography (CT) images were acquired immediately prior to the commencement of the PET scan with the following settings: 100 kVp, 90 mAs regulated using the manufacturer’s dose reduction software, and 3 mm slice thickness. The raw PET data were reconstructed based on the corresponding CT dataset according to the OSEM protocol (2 iterations and 24 subsets with a 6.4 mm Gaussian filter, including time-of-flight, attenuation, and scattering correction, and incorporating the point spread function (SarpIR)).

### 2.3. FMISO PET-MRI Imaging

Within an interval of 10 ± 9 days after the FDG PET exam, patients underwent successive FMISO PET/CT and MRI scans (13 ± 5 min delay) in random order. The exams were performed in the same patient position between the two acquisition systems using a flat table, skin markers, and a specific homemade wedge under the head that allowed precise repositioning. Patients received a bolus intravenous injection of 4 MBq/kg of FMISO. Head and neck PET acquisition began 183 ± 6 min after injection on the previously described PET-CT device, with an imaging time of 4 min per bed position. The corresponding computed tomography (CT) images were acquired immediately before the start of the PET acquisition with the following parameters: 100 kVp, 90 mAs regulated using the manufacturer’s dose reduction software, and 3.75 mm slice thickness. The raw PET data were reconstructed based on the corresponding CT dataset with the BPL algorithm parameter β set to 350, according to Texte et al. [16].

MRI scans were performed on a GE Optima MR450w 1.5 Tesla (T) MRI scanner (General Electric, Milwaukee, WI, USA) using a 20-channel head coil. Multiplanar (axial, coronal, and sagittal) T2-weighted (T2w) and T1-weighted (T1w) MRI sequences were acquired as recommended for diagnostic and target delineation in head and neck cancer [17,18]. DW-MRI, T1 and T2 mapping, and DCE-MRI acquisitions were performed after standard T1w and T2w imaging. The total acquisition time was approximately 30 min for the entire MRI examination. DW-MRI data were acquired using a spin-echo echo-planar imaging sequence (SE-EPI). T1 mapping was obtained by using the method of different flip angles repeated four times (VFA), using a gradient-echo sequence with a gradient spoiler (GE-SPGR), and T2 mapping by a spin-echo sequence with multiple echo times (SE-ME). DW-MRI data were acquired using spin-echo echo-planar imaging (SE-EPI) with different b-values (0, 500, and 1000 s/mm^2^). The apparent diffusion coefficient (ADC) mapping was calculated using ReadyView software (version DV26.0-R03-1831.b, General Electric, Milwaukee, WI, USA). T1 mapping was obtained by using the method of different flip angles repeated four times (VFA) using a gradient-echo sequence with a gradient spoiler (GE-SPGR) and T2 mapping by a spin-echo sequence with multiple echo times (SE-ME). T1 and T2 mappings were calculated using the OleaNova+ module of the Olea Sphere software (version 3.0, OLEA MEDICAL, La Ciotat, France). The dynamic DCE-MRI series images were acquired after injection of a bolus of Dotarem (0.5 mmol/mL gadoteric acid concentration) through an antecubital venous catheter at a rate of 5cc/s followed by a 20 mL saline lavage after the acquisition of 5 images. A total of 48 dynamic images were obtained with a temporal resolution of 10 s/image. Ktrans, KEP, Vp and Ve mappings were calculated using the Permeabilty module of the Olea Sphere software (version 3.0, OLEA MEDICAL, La Ciotat, France). The set of MRI parameters of the sequences used is given in Table 2.

### 2.4. Registration, Segmentation, and Definition of Tumor Volumes as Well as Hypoxic and Normoxic Sub Volumes

FDG PET/CT and FMISO PET/MRI images were transferred to a Dosisoft workstation (v3.1, Oncoplanet, DosiSoft, Cachan, France). All series were co-registered without resampling the voxel size with a rigid block-matching method centered on the tumor. Physicians were allowed to manually correct obvious registration defects.

In the first step, the tumor lesion was segmented on FDG PET images by thresholding 40% of the standardized maximum uptake value [19] to give the metabolic tumor volume (Figure 1(a1)). In the second step, the tumor lesion was segmented from the weighted MRI image set. This segmentation was performed manually by a radiologist with >30 years of experience in head and neck imaging using the recommended MRI sequences for diagnostic imaging to obtain the MRI tumor volume (Figure 1(b1)). Then, a third volume corresponding to the total tumor volume resulting from the association of the tumor volumes obtained via FDG PET and MRI was defined using the Boolean volume union operator (Figure 1(c1)).

The three defined tumor volumes were plotted on FMISO PET images. In each of the three volumes, when possible, three hypoxic subvolumes (HsV) were defined by applying a relative threshold that was 1.4 times the mean SUV (SUVmean) of the sternocleidomastoid muscle contralateral to the lesion [20] to give HsV-FDG (Figure 1(a2)), HsV-MRI (Figure 1(b2)), and HsV_MRIuFDG (Figure 1(c2)), respectively.

In addition, three normoxic subvolumes (NsV) were defined by the Boolean operator of the non-inclusion of hypoxic subvolumes within the three previously defined tumor volumes (Figure 1(a3,b3,c3)).

All the obtained segmentations were propagated and registered to the quantitative MRI series for analyses.

### 2.5. Quantitative Parameters

Tumor volumes were measured using FDG PET (MTV), FMISO PET (HTV), and MRI. Within each segmentation obtained, the parameters of the biodistribution of FDG PET, FMISO PET, and quantitative MRI sequences were performed as follows: for FDG PET, the standardized maximum uptake value (SUVmax) and the mean value (SUVmean) were obtained, and the total lesion glycolysis (TLG) was measured by multiplying the MTV and the SUVmean.

For FMISO-PET, the SUVmax and the SUVmean were obtained, and the tumor-to-muscle ratio (TMR) was found by dividing the SUVmax of the lesion by the SUVmean of the contralateral sternocleidomastoid muscle. Lesions were considered hypoxic when the TMR was greater than 1.25 [5,21].

MRI data analysis was performed for each volume and sub-volumes by measuring the quantitative parameters by the mean DCE-MRI values (Ktrans, Kep, Vp, and Ve), ADC, and mean and maximum T1 and T2 mapping values.

### 2.6. Statistical Analysis

Quantitative MRI parameter measurements were compared between the hypoxic segmentation and the MRI, FDG, and MRIuFDG segmentations. The Dice similarity index, Jaccard similarity index, and overlap fraction were measured between the FDG FDG and MRI segmentations.

To determine the reference hypoxic volume, a Friedman paired-data test was performed to evaluate the three hypoxic volumes (HsV-MRI, HsV-FDG, and HsV_MRIuFDG). In cases of significant differences, pairwise comparison Wilcoxon signed-rank tests were performed.

We compared the parameters measured from the FDG PET, FMISO PET, and MRI between two groups of patients (with and without a measurable hypoxic volume) by using non-parametric Wilcoxon tests for independent data. SUV parameters and volumes were described using mean and standard deviation, median, minimum, and maximum values. Evaluation of the difference in parameters (Ktrans, Kep, Ve, Vp, ADC, T1mapping, and T2mapping) was performed following the different segmentations in both populations by a pairwise comparison Wilcoxon signed-rank test compared to the reference hypoxic volume. The results are given by boxplots. p-values were adjusted for the Bonferroni correction in the case of pairwise comparisons performed in our analyses.

## 3. Results

### 3.1. Quantitative Parameters of PET and MRI

#### 3.1.1. Analysis of PET Data and MRI Volume

For FDG PET, the SUVmax was 12.3 cm^3^ (±6.6), and the volume at a 40% threshold for the SUVmax was 15.3 cm^3^ (±16.9). For FMISO PET, 13 of the 16 lesions were identified as hypoxic with an SUVmax of 2.3 (±0.56). For MRI, the measured volume was 35.3 cm3 (±43.2). Tumor volumes determined by FDG PET were smaller than those determined by MRI. The Dice, Jaccard, and overlap fraction similarity indices between the lesion volumes measured via FDG PET and MRI were 0.43, 0.28 and 0.71, respectively. These results show that the FDG PET tumor volume is only partially contained in the MRI tumor volume. The hypoxic sub-volumes were 1.4 cm^3^ (±1.8) from the HsV-FDG segmentation, 2.6 cm^3^ (±5.2) from the HsV-MRI segmentation, and 3.3 cm3 (±6.4) from the HsV-MRIuFDG segmentation (Table 3 and Figure 2).

#### 3.1.2. Comparison of Hypoxic Sub-Volumes

The overall difference in the three hypoxic sub-volumes was assessed by comparing a statistically significant difference between the HsV-MRIuFDG volume and the other two volumes of HsV-FDG and HsV-MRI (Figure 3). We did not find a statistically significant difference between the HsV-FDG and HsV-MRI volumes. The HsV-MRIuFDG segmentation was considered as the reference hypoxic sub-volume in our analyses.

#### 3.1.3. Comparison of PET and MRI Parameters between Patients with and without Measurable Hypoxic Volumes

Quantitative FDG PET and FMISO PET parameters were compared between patients with (*n* = 13) and without (*n* = 3) measurable hypoxic volumes. Non-parametric Wilcoxon tests for unpaired data were performed (Table 4 and Figure A1). The results show statistically significant differences between patients with and without measurable hypoxic lesions based on the SUVmax, SUVmean, and SUVpeak of the FDG PET, as well as the SUV max of the FMISO PET. In contrast, we found no statistically significant difference between the two groups of patients in measured FDG PET and MRI tumor volumes and quantitative MRI parameters.

### 3.2. Quantitative MRI Parameters According to Hypoxic Segmentation

The quantitative MRI parameters between the different normoxic volumes (MRI, FDG, and MRI-FDG) and the hypoxic HsV-MRIuFDG sub-volume were compared via non-parametric Wilcoxon tests for the 13 patients with a hypoxic lesion. The results show statistically significant differences between hypoxic and normoxic volumes for the parameters ADC (*p* = 0.01), T1 mapping max (*p* = 0.01), and T2 mapping max (*p* = 0.01) (Figure 4 and Figure A2). ADC values tended to be higher in the hypoxic volumes than in the normoxic volumes. Conversely, T1 and T2 max mapping values are lower in the hypoxic volume. The other parameters showed no statistically significant differences between the hypoxic subvolume and the normoxic volumes.

## 4. Discussion

Although several studies have highlighted the complementary nature of PET and MRI for radiotherapy planning in head and neck cancers, this is, to our knowledge, the first study showing the potential value of combining tumor volumes obtained via PET and MRI to define the hypoxic lesion’s subvolume [11,22]

It can be seen that our study suffers from a lack of patients. Only 16 patients were included despite the 38 pre-inclusions performed. Defects in the production of 18F-FMISO and the COVID-19 crisis limited the number of inclusions. The vast majority of the patients included had hypoxic tumors (13 patients out of the 16 included).

The hypoxic volume is smaller than the metabolic volume. This result is comparable to that from the work of our team recently published within the framework of the RTEP6 protocol [23]. Indeed, in 20 patients with lung cancer, it was shown that the hypoxic volume/metabolic volume ratio was, on average, 12% (compared to 10% in our study). A recent study also found that the majority of patients had hypoxic tumors but that the hypoxic volume was much lower than the metabolic volume (between 7 and 25% lower). In this previous work, the dose of radiotherapy could be increased by 10% with a constant spread and without a significant increase in toxicity. This dose increase seemed to be related to better control of the cancer [24].

Regarding tumor volume, the comparison of volumes defined from FDG PET and those with weighted MRI sequences shows that they are different with a statistically larger MRI volume.

Only some quantitative parameters in PET imaging showed statistically significant differences between the two populations (with and without measurable hypoxic volumes): SUVmax, SUVmean, and SUVpeak for FDG PET and SUVmax for FMISO PET. These parameters could be used to characterize the presence of hypoxia. Although the number of patients in the population without a measurable hypoxia volume is considerably low in our study, these results had already been highlighted [23,25,26], showing that the presence of hypoxia is associated with an elevated SUVmax in FDG PET. These results confirm that these two radiotracers provide different but complementary information [23].

The results of our study show statistically significant differences regarding quantitative MRI of free water diffusion and relaxometry techniques between hypoxic and normoxic volumes. These results regarding diffusion MRI are comparable to the preclinical results from others studies that have correlated the presence of hypoxia with ADC mapping [27,28]. In a comparable study, including 21 patients with a hypoxic head and neck cancer lesion who were imaged before and during radiotherapy (at 2 and at 5 weeks), the author found significantly lower ADC values in the hypoxic subvolume compared with the normoxic subvolume, which is opposite to our results, where ADC was significantly higher before treatment [11]. Our results could be explained by the changes in the cellular structures of the tumor, such as cell membrane rupture, that may occur due to a process of apoptotic cell death and would lead to a decrease in the cellularity of the tumor and, thus, an increase in ADC values in these regions [29]. Recently, in a study investigating the correlation between the tumor microenvironment, proliferation, and tumor hypoxia by comparing ADC measurements in 20 patients with oropharyngeal squamous cell carcinoma, Swartz et al. [30] showed high ADC values in hypoxic lesions, but unfortunately, without observing a correlation with the HIF-1 protein expression.

Relaxometry techniques allow for the true intrinsic measurement of the tissue’s T1 and T2 parameters. This approach allowed for tissue and pathology characterization [31]. Following recent work by our team to optimize the measurement of these relaxometry techniques for an acceptable clinical time [32], we hypothesized in this work that a difference in the T1 and T2 measurements could be observed between normoxic and hypoxic tissue. However, to our knowledge, no clinical study has explored the potential interest of these relaxometry techniques to demonstrate tumor hypoxia. Our results show significantly lower T1 and T2 values of hypoxic intratumoral tissue compared to normoxic tissues underlying their potential ability to characterize tissue oxygenation. The physico-chemical environment has an influence on the T1 and T2 relaxation times of tissues in MRI. In particular, oxyhemoglobin in oxygenated red blood cells is a diamagnetic molecule, where deoxyhemoglobin in deoxygenated red blood cells is paramagnetic [14]. An increase in deoxyhemoglobin concentration leads to an acceleration of spin-lattice relaxation rates and, thus, a decrease in T1 and T2 relaxation times.

In a preclinical study, Serša et al. [13] showed that a model can be used for the calculation of the predicted hypoxic level map based on the apparent diffusion coefficient measured by magnetic resonance and T2 maps. Our future immunohistochemical analyses of tumor slides should shed light, on a patient-by-patient basis, on the biological phenomenon behind the differences in ADC measurements and T1 and T2 maps between hypoxic and normoxic tissues and propose models combining these quantitative techniques. Additionally, it would be interesting to study the predictive and prognostic value of ADC, T1max, and T2max.

In our analyses, we considered voxels with the maximum value only for T1 and T2 mapping. The use of an average value for all voxels in a subvolume may not be sufficiently reliable because the intra-tumor heterogeneity [33] may explain the lack of significance of our results regarding the average T1 and T2 values (Figure A2). Maximum and minimum values for diffusion MRI and DCE-MRI were not considered as usable because of a high noise level. This high noise level was limited by the increased voxel size in diffusion MRI and DCE-MRI, limiting the anatomical accuracy of the measurement.

There was no significant difference regarding the parameters extracted from perfusion MRI and the perfusion parameter ktrans between the hypoxic and normoxic subvolumes before treatment. This lack of difference is probably secondarily due to our small number of patients. Nevertheless, our results show that DCE-MRI seems to be less relevant for the study of hypoxia than diffusion MRI and T1 and T2 mapping measurements. The demonstration of perfusion variations is complex because many factors can influence these variation, such as the histological type of the tumor, the vascular architecture or the tumor size. DCE-MRI may reflect indirect estimates of tumor hypoxia and, in some circumstances, may not reflect hypoxia [34].

To our knowledge, there is no single software used in clinical routines to perform the post-processing of acquisitions and the registration of a series for radiotherapy treatment planning with the possibility of segmenting in the DICOM RT format. In particular, the registration process requires interpolation and, therefore, resampling of the target image matrix to the spatial resolution of the reference image. This operation modifies the original values of the voxels of the newly registered target image, which has an impact on the quantitative measurements of the image. The specificity of the software that we used in our study offers the possibility to perform a quantitative measurement of the original voxels of the target image, even when it has been registered on a reference image. Consequently, the measurements we performed on the re-registered image series are not biased by the interpolation process performed during the re-registration. Additionally, Complementary techniques, such as semi-automatic segmentation of quantitative MRI images, should be developed to improve the definition of hypoxic subvolumes in future studies and limit observer variability.

## 5. Conclusions

This study confirms the important prevalence of hypoxia in head and neck cancers but also demonstrates that the hypoxic volume is low compared to the metabolic volumes. Quantitative MRI, based on free water diffusion and T1 and T2 mapping, seems to be able to identify intra-tumoral hypoxic sub-volumes for additional radiotherapy doses. These results must be confirmed by more studies, and clinical trials should be performed to support these preliminary data.

## Figures and Tables

**Figure 1 cancers-15-01918-f001:**
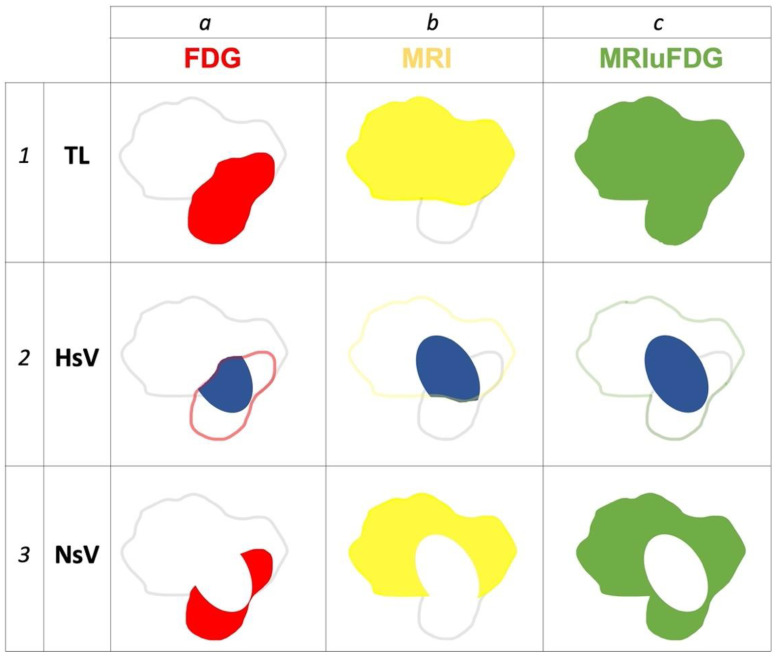
Example of different segmentations obtained from a patient included in the RTEP8-Hyponeck study. Columns a, b, and c represent FDG PET (red), MRI (yellow), and the union of MRI and FDG PET (green), respectively. Lines 1, 2, and 3 represent the segmentations of the tumor lesion (TL; solid color), hypoxic (HsV; blue color), and normoxic (NsV; white color) subvolumes.

**Figure 2 cancers-15-01918-f002:**
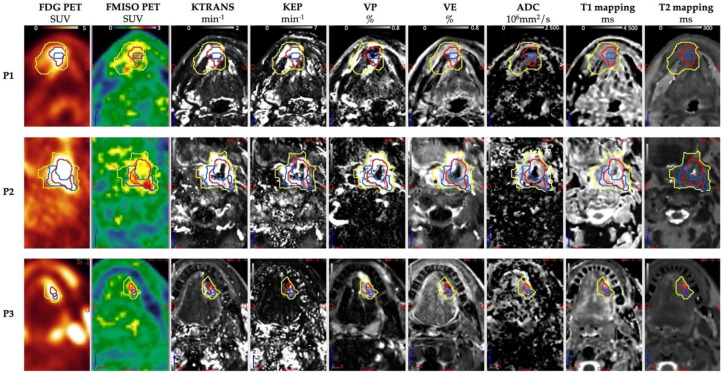
Multimodal PET/CT MRI images obtained from patients 1, 2, and 14 of the RETP8 population. For more details, see patient characteristics in Table 1. MRI tumor segmentation in yellow, FDG PET in red, and FMISO PET hypoxic subvolume in blue on each quantitative PET and MRI series.

**Figure 3 cancers-15-01918-f003:**
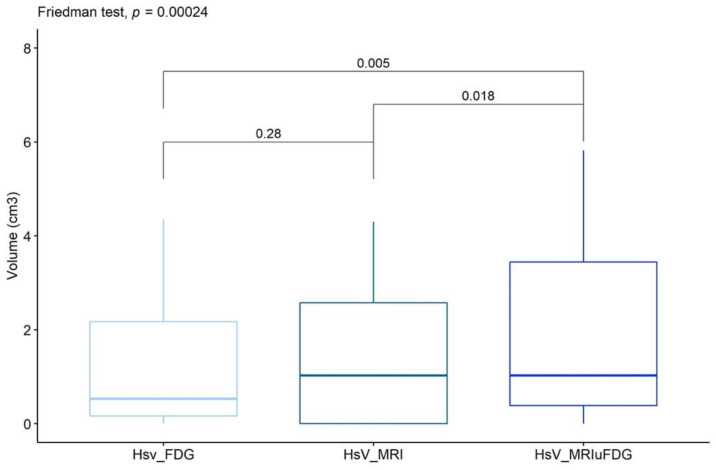
Wilcoxon signed-rank tests between the hypoxic sub volumes HsV_FDG (light blue), HsV_MRI (south sea blue), and HsV_MRIuFDG (dark blue).

**Figure 4 cancers-15-01918-f004:**
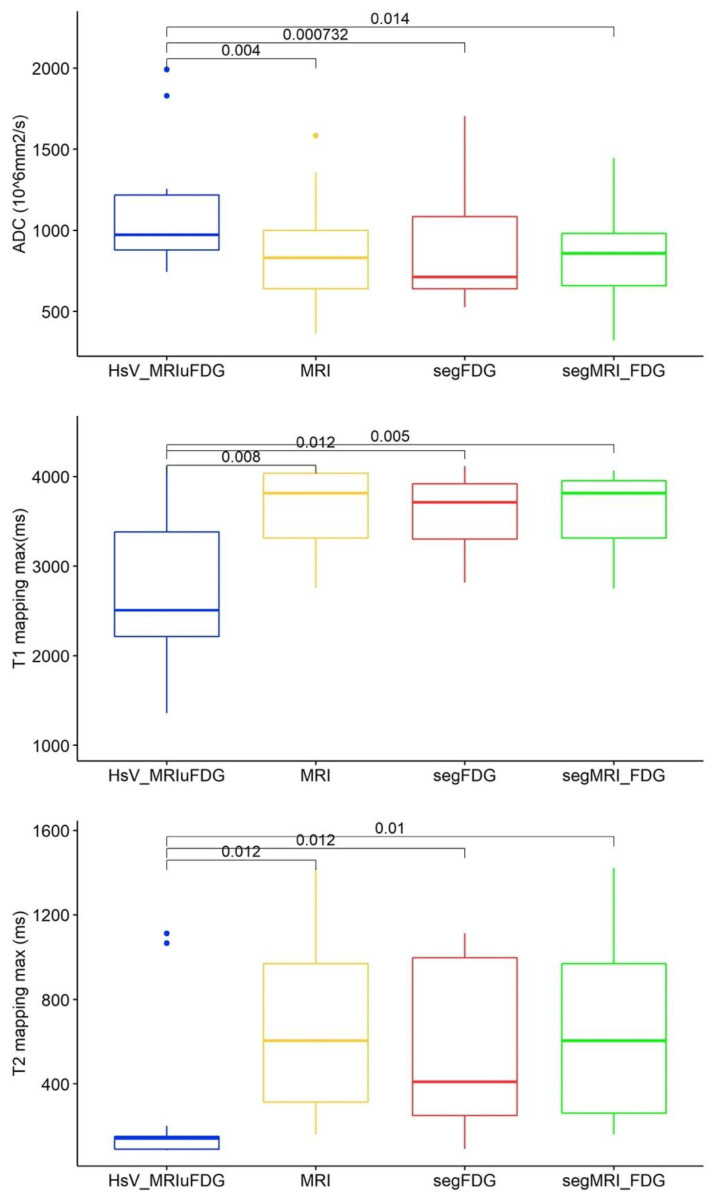
Pairwise Wilcoxon signed-rank tests of quantitative MRI parameters in patients with an identified hypoxic lesion (*n* = 13). The reference hypoxic HsV_MRIuFDG segmentation is shown in blue, and the normoxic MRI, FDG PET and MRIuFDG segmentations are shown in yellow, red and green, respectively.

**Table 1 cancers-15-01918-t001:** Patient characteristics.

Patient	Sex	Age	T-Stage	HPV Statut	T-Site
1	H	64	T2N3b	Positive	Oral cavity
2	H	56	T4aN0	Negative	Oral cavity
3	H	63	T4aN0	Negative	Oral cavity
4	H	76	T2N0	Negative	Oropharynx
5	H	60	T3N0	Negative	Oropharynx
6	H	72	T3N1	Negative	Oropharynx
7	H	59	T4aNx	Negative	Oral cavity
8	H	35	T2N3b	Negative	Oral cavity
9	H	54	T4aN2b	Negative	Oropharynx
10	H	70	T4aN0	Negative	Oral cavity
11	H	78	T3N0	Negative	Oral cavity
12	H	70	T4aN3b	Negative	Oral cavity
13	H	77	T4aN2b	Negative	Oropharynx
14	H	54	T4aN1	Negative	Oral cavity
15	F	75	T4aN3b	Negative	Oral cavity
16	F	64	T4aN0	Negative	Oral cavity

**Table 2 cancers-15-01918-t002:** Acquistion parameters for MRI.

Settings	DCE-MRI	T1 Mapping	T2 Mapping	DWI
**Plane**	Axial	Axial	Axial	Axial
**TR (ms)**	8.4	15	1000	8091
**TE (ms)**	3.1	3	5.98–11.96–17.94–23.92–29.9–35.88–41.86–47.84	80.5
**Flip Angle (degrees)**	25	3–10–20–30	90	90
**Matrix size (pixels)**	256 × 256	128 × 128	128 × 128	256 × 256
**Slice thickness (mm)**	3.4	2	2	3
**Slice spacing (mm)**	0	0	0	0.3
**Pixel size (mm)**	0.9 × 0.9	1.01 × 1.00	1.01 × 1.01	0.94 × 0.94

**Table 3 cancers-15-01918-t003:** Quantitative parameters of SUV and MRI volumes extracted from FDG PET, FMISO PET and MRI. * represents patients (*n* = 3) whose tumor lesion does not contain a hypoxia sub-volume.

Patients	FDG SUVmax	MTV cm^3^	TLG	FMISO SUVmax	HSv FDG cm^3^	HSv MRI cm^3^	HSv MRIuFDG cm^3^	MRI Volume cm^3^
1 *	8.4	2.2	10.78	2.64	0.0	0.0	0.0	2.63
2	15.4	71.54	676.77	2.08	2.03	2.25	2.25	162.89
3	10.43	13.42	77.43	2.51	0.86	1.22	1.22	29.1
4	13.32	7.82	66.08	2.24	1.2	1.76	2.22	10.92
5	14.67	18.75	204.19	2.46	2.59	4.3	4.84	49.96
6	10.87	11.86	92.98	2.12	0.2	0.2	0.46	10.48
7	10.86	17.29	135.21	1.75	0.17	0.83	0.83	59.19
8	11.32	4.4	26.62	2.79	0.66	1.71	2.98	7.01
9 *	8.69	9.76	47.43	1.25	0.0	0.0	0.0	30.51
10	11.14	31.59	219.87	2.44	4.35	4.13	5.82	62.22
11 *	7.81	1.8	8.17	1.92	0.0	0.0	0.0	1.33
12	5.74	12.22	48.02	2.3	0.4	0.0	0.51	6.72
13	14.18	20.12	179.47	3.56	5.75	21.49	26.16	93.59
14	9.52	7.09	38.71	1.84	0.37	0.76	0.76	15.49
15	34.9	12.23	237.87	3.26	4.01	3.55	4.91	21.8
16	9.25	2.01	15.74	2.28	0.15	0.0	0.15	1.76
Mean	12.3	15.3	130.3	2.3	1.4	2.6	3.3	35.3
Median (q1; q3)	10.9 (9.1; 13.5)	12.1 (6.4; 17.6)	71.8 (35.7; 185.6)	2.3 (2; 2.5)	0.5 (0.2; 2.2)	1 (0; 2.6)	1 (0.4; 3.4)	18.6 (6.9; 52.3)
min; max	5.7; 34.9	1.8; 71.5	8.2; 676.7	1.2; 3.6	0; 5.7	0; 21.5	0; 26.2	1.3; 162.9

**Table 4 cancers-15-01918-t004:** Non-parametric Wilcoxon tests for independent data between patients with and without measurable hypoxic volumes.

	Hypoxic Volume not Measurable (*n* = 3)	Hypoxic Volume Measurable (*n* = 13)	*p*-Value
**FDG PET**			
**SUV max**			**0.01**
mean (±standard deviation)	8.7 (±2.0)	14.4 (±7.5)	
median (q1;q3)	8.5 (8.0; 10.3)	12.3 (10.6; 14.5)	
min; max	5.7; 10.9	9.2; 34.9	
**SUV mean**			**0.02**
mean (± standard deviation)	5.6 (±1.7)	8.9 (±4.1)	
median (q1;q3)	4.9 (4.6; 7.1)	8.1 (6.3; 9.3)	
min; max	3.9; 7.8	5.5; 19.4	
**SUV peak**			**0.05**
n (NA)	4 (2)	8 (2)	
mean (±standard deviation)	7.3 (±2.2)	12.7 (±5.6)	
median (q1;q3)	7.6 (5.9; 9.0)	11.6 (9.6; 13.3)	
min; max	4.7; 9.3	7.6; 25.6	
**MTV**			0.26
mean (±standard deviation)	9.2 (±6.1)	18.9 (±20.5)	
median (q1;q3)	10.8 (4.1; 12.1)	12.8 (7.3; 19.8)	
min; max	1.8; 17.3	2.0; 71.5	
**TLG**			0.14
mean (±standard deviation)	57.1 (±49.2)	174.3 (±196.1)	
median (q1;q3)	47.7 (19.9; 81.7)	128.4 (45.6; 215.9)	
min; max	8.2; 135.2	15.7; 676.8	
**FMISO PET**			
**Suv max**			**0.05**
mean (±standard deviation)	2.0 (±0.5)	2.5 (±0.5)	
median (q1;q3]	2.0 (1.8; 2.3]	2.5 (2.2; 2.7]	
min; max	1.2; 2.6	1.8; 3.6	
**MRI**			
**Volume**			0.18
mean (±standard deviation)	18.5 (±22.6)	45.5 (±50.2)	
median (q1;q3]	8.6 (3.6; 25.5]	25.5 (12.1; 59.1]	
min; max	1.3; 59.2	1.8; 162.9	

## Data Availability

The data that support the findings of this study are available from the corresponding author upon reasonable request.

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
