# Peer review of "Quantitative MRI to Characterize Hypoxic Tumors in Comparison to FMISO PET/CT for Radiotherapy in Oropharynx Cancers"

_cancers, 2023, doi:10.3390/cancers15061918_

Round 1
Reviewer 1 Report
I would suggest that Colleagues increase the statistics and experiment with the method with other head and neck cancers.
I find the work particularly appreciable.
I have a question for them the use of numerous software in the study how did it affect the overall evaluation of the data?
What knowledge is required for a nuclear physician of these softwares to understand their limitations and likely bugs in the evaluations?
Author Response
Responses to Reviewer 1
I would suggest that Colleagues increase the statistics and experiment with the method with other head and neck cancers.
The major limitation of this work is indeed the small number of patients. The COVID-19 health crisis occurred shortly after the first two inclusions. The production of 18F-FMISO was therefore stopped by the laboratory for 266 days, preventing us from including patients over this period. In addition, the PET/CT machine on which the study started had to be replaced which necessitated stopping the inclusions because of the important technological gap with the new machine (SiPM) which could lead to heterogenous data from first included patients with the previous system and those included with the current one. This work will be continued through new projects with its new technologies in order to support and/or confirm the results obtained.
To prevent some weak significant results leading to multiplicity of tests with pairwise comparisons tests, we considered a correction of Bonferroni. This point was not mentionned in the part of statistical analysis in original manuscript whereas information is important. We added it in the revised manuscript.
We are well aware of the small number of included patients in our study could also have consequences to obtain non significant results, thus we performed some post hoc power calculations to have an idea of the number of patients to include for future studies. We consider the observed proportion of patients with hypoxic and non hypoxic tumors (3/16 vs 13/16), the observed means and associated standard deviations for different parameters of interest. Since there is not only one parameter of interest in our study, it was difficult to make one global power calculation. For these calculations, we consider Kep and T2 as potential candidates parameters for which significant results would also be expected. By considering a bilateral formulation of Student test with alpha risk of 5%, we would be able to have a probability more than 90% to show a significant mean difference of 0.49 min-1 for Kep associated to a standard deviation of 0.23 between 24 patients with hypoxic tumors and 6 with non hypoxic tumors. With similar hypothesis, we would have a probability close to 80% to put in evidence a significant mean difference of 8.7 ms (std deviation of 6.6) for parameter T2 with 30 patients (6 vs 24).
I find the work particularly appreciable.
I have a question for them the use of numerous software in the study how did it affect the overall evaluation of the data?
We thank the reviewer for this comment. We add this point to the discussion. « To our knowledge, there is no single software used in clinical routine to perform post-processing of acquisitions, registration of series for radiotherapy treatment planning with the possibility of segmenting in DICOM RT format. In particular the registration process which requires interpolation and therefore resampling of the target image matrix to the spatial resolution of the reference image. This operation modifies the original values of the voxels of the newly registered target image, which has an impact on the quantitative measurements of the image. The specificity of software that we used in our study, offers the possibility to perform a quantitative measurement of the original voxels of the target image, even when it has been registered on a reference image. Consequently, the measurements we performed on the re-registered image series are not biased by the interpolation process performed during the re-registration. »
What knowledge is required for a nuclear physician of these softwares to understand their limitations and likely bugs in the evaluations?
The nuclear medicine physician should be aware of the results of the evaluations made of these software programs during the tests carried out before the use of these data/evaluations. As this type of data is to be used for radiotherapy treatment, a process of acquisition parameters and image quality control (from signal detection to the final image obtained) is necessary and recommended (Puspasari et al, IJRI, 2022 ; Gouel et al, Frontiers in Oncol, 2022 ; Decazes et al, Frontiers in Oncol, 2021 ; Kurz et al, Radiation Oncol, 2020 ; Johnstone et al, British Journal of Radiology, 2020 ; Thureau et al, Radioation Oncol, 2018 ; Decazes el al, QJNNMI, 2018)
We have used the MDPI services editing process in English before sending you a new version.
Reviewer 2 Report
I suggest minor english review; your research is very interesting and results are in accordance to knowledge confirming the importance of hypoxic volumes in treatment outcome; limit of the present paper is the small number of patients which forbids final conclusions. I would recommend to continue this study in order to have stronger results
Author Response
Responses to Reviewer 2
I suggest minor english review; your research is very interesting and results are in accordance to knowledge confirming the importance of hypoxic volumes in treatment outcome; limit of the present paper is the small number of patients which forbids final conclusions. I would recommend to continue this study in order to have stronger results.
We thank the reviewer for his evaluation and comments.
We have used the MDPI services editing process in English before sending you a new version.
The major limitation of this work is indeed the small number of patients. The COVID-19 health crisis occurred shortly after the first two inclusions. The production of 18F-FMISO was therefore stopped by the laboratory for 266 days, preventing us from including patients over this period. In addition, the PET/CT machine on which the study started had to be replaced which necessitated stopping the inclusions because of the important technological gap with the new machine (SiPM). This work will be continued through new projects with its new technologies in order to support and/or confirm the results obtained.
Reviewer 3 Report
The authors report a study showing the value of combining tumor volumes obtained on PET and MRI to define the hypoxic lesion subvolume. Quantitative MRI parameters ADC, T1 mapping, and T2 mapping showed differences between hypoxic and normoxic volumes. The objective of this work is to define, from both PET hypoxia data (FMISO) and MRI the hypoxic volume for which there would be an interest to increase the radiotherapy dose in head and neck cancers. The data are interesting and the manuscript is original and well written. The manuscript may be accepted in the form in which it was submitted.
Author Response
Responses to Reviewer 3
The authors report a study showing the value of combining tumor volumes obtained on PET and MRI to define the hypoxic lesion subvolume. Quantitative MRI parameters ADC, T1 mapping, and T2 mapping showed differences between hypoxic and normoxic volumes. The objective of this work is to define, from both PET hypoxia data (FMISO) and MRI the hypoxic volume for which there would be an interest to increase the radiotherapy dose in head and neck cancers. The data are interesting and the manuscript is original and well written. The manuscript may be accepted in the form in which it was submitted.
We thank the reviewer for his evaluation. We have used the MDPI services editing process in English before sending you a new version.
Round 2
Reviewer 2 Report
You correctly modified your paper according to indications and in my opinion is now worth of publication